# Early Atf4 activity drives airway club and goblet cell differentiation

Juan F Barrera-Lopez, Guadalupe Cumplido-Laso, Marcos Olivera-Gomez ⓘ, Sergio Garrido-Jimenez, Selene Diaz-Chamorro, Clara M Mateos-Quiros, Dixan A Benitez, Francisco Centeno ⓘ, Sonia Mulero-Navarro ⓘ, Angel C Roman, Jose M Carvajal-Gonzalez ⓘ

**Activating transcription factor 4 (Atf4), which is modulated by the protein kinase RNA-like ER kinase (PERK), is a stress-induced transcription factor responsible for controlling the expression of a wide range of adaptive genes, enabling cells to withstand stressful conditions. However, the impact of the Atf4 signaling pathway on airway regeneration remains poorly understood. In this study, we used mouse airway epithelial cell culture models to investigate the role of PERK/Atf4 in respiratory tract differentiation. Through pharmacological inhibition and silencing of ATF4, we uncovered the crucial involvement of PERK/Atf4 in the differentiation of basal stem cells, leading to a reduction in the number of secretory cells. ChIP-seq analysis revealed direct binding of ATF4 to regulatory elements of genes associated with osteoblast differentiation and secretory cell function. Our findings provide valuable insights into the role of ATF4 in airway epithelial differentiation and its potential involvement in innate immune responses and cellular adaptation to stress.**

## Introduction

The respiratory tract is lined by a single layer of epithelial cells, which serves as a crucial barrier separating our internal body from the external environment (Hewitt & Lloyd, 2021). To ensure the proper functioning of the respiratory tract, including its ability to regenerate in response to environmental exposures, it is important to comprehend and regulate the processes of differentiation and self-renewal in the adult stem cells present within the epithelium (Whitsett & Alenghat, 2015; Davis & Wypych, 2021). Moreover, gaining a deeper understanding of the mechanisms underlying epithelial development is essential for the development of more comprehensive and innovative experimental model systems, encompassing both two-dimensional and three-dimensional culture settings (Pampaloni et al, 2007; Drost & Clevers, 2018).

At the steady state, the airway epithelium consists of various specialized cell types (Hewitt & Lloyd, 2021). Among these cell types, the major constituents of the airway epithelium are basal stem cells (BSCs), secretory cells (SCs), and multiciliated cells (MCCs) (Cardoso, 2001; Rawlins & Hogan, 2006). BSCs are multipotent stem cells responsible for tissue regeneration, exhibiting the ability to fully regenerate the epithelium (Rock et al, 2009; Yang et al, 2018). SCs, including club cells (CCs) and goblet cells (GCs), produce mucus essential for trapping pathogens and pollutants (Rawlins et al, 2009). In addition, MCCs play a crucial role in continuously propelling mucus toward the mouth, thereby keeping the airway tract clear (Bustamante-Marin & Ostrowski, 2017). Furthermore, two intermediate cell states have been described. Deuterosomal cells, which are precursor cells for MCCs, express genes associated with centriole amplification, such as *Foxn4*, *Cdc20b*, and *Deup1* (Revinski et al, 2018; García et al, 2019). Suprabasal cells, intermediate between basal cells and CCs, exhibit lower expression levels of p63 compared with basal cells and migration of the nucleus toward more apical planes within the epithelium (Wu et al, 2022). Apart from these three/six cell types, other less abundant cell types (~1% of cells) are present in the airway epithelium in vivo. These include Tuft cells involved in immunity, pulmonary neuroendocrine cells acting as environmental sensors, and pulmonary ionocytes (Montoro et al, 2018; Plasschaert et al, 2018), which were recently identified using single-cell RNA-sequencing technology. Understanding the molecular mechanisms underlying this cellular complexity necessitates detailed analysis of the transcriptional programs specific to each cell type.

As mentioned earlier, BSCs are primarily responsible for tissue regeneration (Rock et al, 2010; Wu et al, 2022), although it has been observed that a subset of SCs (CCs) can undergo dedifferentiation to BSCs and subsequently contribute to epithelial regeneration (Rawlins et al, 2009; Tata et al, 2013). Broadly speaking, there are two main routes of differentiation originating from BSCs. One involves the differentiation of BSCs into MCCs, whereas the other involves their differentiation into SCs. Activation of specific transcription factors has been associated with the in vivo development

Departamento de Bioquímica, Biología Molecular y Genética, Facultad de Ciencias, Universidad de Extremadura, Badajoz, Spain

Correspondence: jmcarvaj@unex.es

and regeneration, as well as the in vitro dedifferentiation, trans-differentiation, and differentiation of precursor cells into more specialized and tissue-specific cell types (Jopling et al, 2011; Merrell & Stanger, 2016). Although certain transcription factors, such as p53, have been identified as pivotal players in the differentiation of BSCs into both SCs and MCCs (Garrido-Jimenez et al, 2021), it is evident that there are also distinct transcriptional regulatory mechanisms spe-cific to each cell type. Numerous MCC-specific transcription factors have been identified, including Mcidas, Foxj1, Myb, Rfx2, and Rfx3 in the airway epithelium, as well as MCCs from other tissues. These transcription factors work together to coordinate the differentiation and maturation of MCCs in the airway epithelium, ensuring the proper formation and functioning of motile cilia. For example, it is well established that Mcidas is essential for activating Foxj1 and subsequent gene expression activates the expression of genes in-volved in ciliogenesis, axoneme assembly, and motility (Look et al, 2001; Yu et al, 2008; Stubbs et al, 2012; Choksi et al, 2014). In general, down-regulation of these transcription factors specifically leads to a decrease in the population of MCCs both in vitro and in vivo. On the contrary, the process of SC differentiation is also a highly intricate phenomenon, encompassing the transition from a non-secretory to a secretory phenotype. The precise orchestration of this process ne-cessitates the interplay and cooperation of multiple transcription factors. Although several transcription factors have been implicated in the regulation of the SC (mostly goblet cell) transcriptional pro-gram, including Spdef, Runx2, Foxa2, Klf5, and Klf4, our understanding of the molecular mechanisms governing SCs, club and goblet, dif-ferentiation remains limited. So far, Spdef (SAM Pointed Domain Containing ETS Transcription Factor) is the best-known transcription factor directly associated with SC differentiation, more specifically goblet cells (Park et al, 2007). In addition, few signaling pathways such as Notch or IL-13 have been closely linked to SC differentiation (Kim et al, 2002; Tsao et al, 2009). Remarkably, IL-13 treatment of airway epithelial cells is one of the current cell culture models used to induce goblet cell hyperplasia (Grünig et al, 1998; Wills-Karp et al, 1998).

In this study, our initial objective was to identify transcription factors (TFs) and their associated protein networks that may play a role in the differentiation of BSCs into MCCs and/or SCs. Through our investigation, we successfully connected the integrated stress response (ISR) pathway involving Perk and its associated TF Atf4 with the specific differentiation of SCs from BSCs.

## Results and Discussion

### The expression of Atf4 is essential for the differentiation of SCs

The recent development of single-cell RNA-sequencing technology has provided a wealth of data that is instrumental in linking proteins and their functions. In this study, we focused on TFs as a protein class to explore novel transcriptional programs relevant to airway epithelium biology. We identified the top 100 TFs expressed in at least one of the three major classes of airway epithelial cells: BSCs, MCCs, and SCs based on TF-related gene ontology (GO) terms (sequence-specific DNA binding, GO:0043565)

(Fig 1A). Subsequently, we assessed the differential gene expression of the top 250 interactors for each TF protein across the three major cell classes in the airway epithelium (Fig 1A). This approach aimed to identify signaling pathways associated with TFs that may have distinct roles in different cell types.

Through the use of our bioinformatic approach, we successfully identified 11 differentially expressed pathways/interactomes of TFs among BSCs, SCs, and MCCs (Fig 1B and C). These pathways included Foxj1, Atf4, Trp53, Irf1, Elf3, Irf3, Epas1, Irf2, Xbp1, Creb3, and Egr1 interactomes. Notably, previous research has demonstrated the crucial involvement of Foxj1 and Trp53 pathways in airway epithelial differentiation (You et al, 2004; Garrido-Jimenez et al, 2021), thereby highlighting the potential impact of other signaling pathways in this differentiation process. To further evaluate these TFs, we examined their connections using the STRING database. As depicted in Fig S1A, we identified two hubs: one comprising Trp53 along with Egr1, Irf1, Irf2, and Irf3, and a second hub consisting of Atf4, Xbp1, and Creb3. However, we did not observe any documented connections be-tween these two hubs or with Foxj1 in the STRING database with a confidence value of 0.9 (Fig S1A).

Subsequently, we assessed the mRNA expression levels of each TF candidate throughout the differentiation process of BSCs using mouse tracheal epithelial cell (MTEC) cultures in an air–liquid in-terface system (see the Materials and Methods section for details). This cell culture method allowed us to easily monitor the differ-entiation of BSCs into SCs and MCCs by examining the expression levels of cell-type markers such as Krt5 for BSCs, Scgb1a1 for SCs, and Foxj1 for MCCs over a period of 14 d (Fig S1B–D). Under these cell culture conditions, we observed that Atf4, Xbp1, and Irf2 TFs maintained consistent expression levels on days 2, 4, 7, and 14 (Fig S1E–M). The significance of this result, combined with several studies that have established a connection between Atf4 and skeletal muscle, as well as osteoblast differentiation, compels us to further explore the Atf4 pathway in our investigations (De Angelis et al, 2003; Yang et al, 2004; Yang & Karsenty, 2004; Xiao et al, 2005; Dobreva et al, 2006).

Atf4 belongs to the ATF/CREB family of basic region/leucine zipper TFs, which can function as either transcription activators or repressors. It serves as a master regulator for the ER stress re-sponse (Pakos-Zebrucka et al, 2016). However, the role of ATF4 in airway epithelial differentiation remains to be explored. To in-vestigate the role of ATF4 in the differentiation process of BSCs to MCCs and SCs, we conducted knockdown experiments targeting Atf4 (ATF4-KD) in BSCs using three different shRNA sequences against Atf4 (Fig 1D and E). Our findings revealed that ATF4-KD did not affect BSC proliferation, as measured by doubling time during cell culture expansion or BSC self-renewal in airway organoid formation (Fig S2A–D). Importantly, ATF4-KD cells were able to form an epithelial monolayer with normal barrier function compared with control conditions (LUC-KD, luciferase knockdown) (Fig 1F).

Next, we proceeded to investigate the potential impact of ATF4 knockdown (ATF4-KD) on the differentiation of BSCs into various airway cell types. To evaluate this, we examined the expression of gene markers specific to BSCs, MCCs, and SCs in ATF4-KD and LUC-KD epithelial monolayers after 14 d of differentiation, employing RT–qPCR analysis. As anticipated, the levels of Krt5 and Trp63,

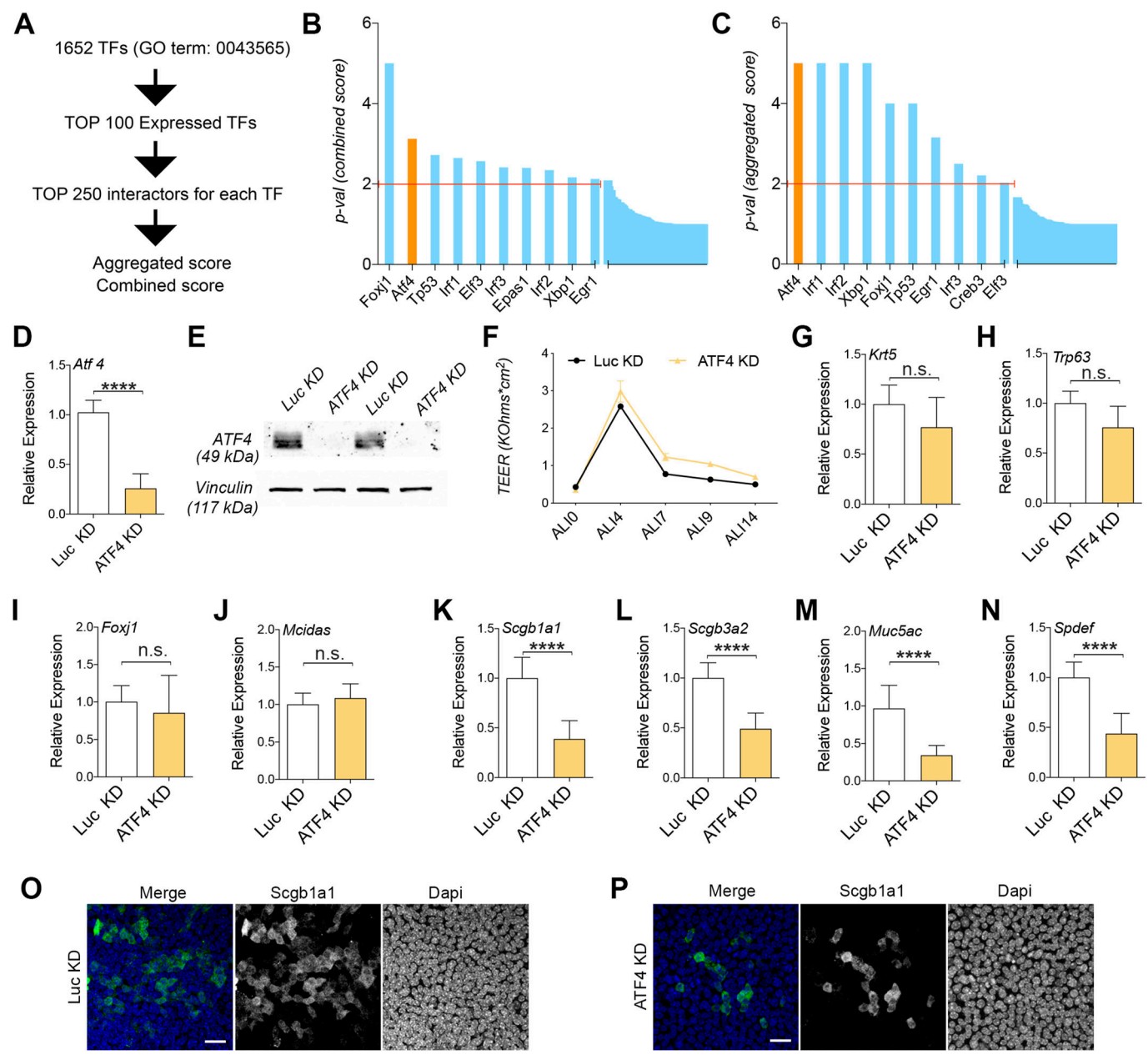

**Figure 1. Atf4 expression is required for SC differentiation.**
**(A)** Workflow of the in silico analysis developed. 1,652 initial proteins with annotated GO:0043565 (sequence-specific DNA binding) were analyzed for detecting interactors with a high score related to airway differentiation. Values for expression were extracted from the accession code GSE102580. **(B)** Histogram of the *P*-values of the combined score (mean value) obtained for candidate transcription factors. **(C)** Histogram of the *P*-values of the aggregated score (sum value) obtained for candidate transcription factors. **(D)** Mean mRNA expression levels of Atf4 assessed in differentiated MTECs infected with control viruses (Luc-KD) or Atf4-shRNAs (ATF4-KD). **(E)** Western blots for ATF4 and vinculin (as a loading control) to determine protein expression levels in differentiated MTECs infected with control viruses (Luc-KD) or Atf4-shRNAs (ATF4-KD). **(F)** Transepithelial resistance (TEER) measure was used to test tight junction permeability during the differentiation process of MTECs in ATF4-KD compared with Luc-KD cells. **(G, H, I, J, K, L, M, N)** mRNA expression levels in Luc versus ATF4-KD. Krt5 and Trp63 for basal cells (G, H), Foxj1 and Mcidas for MCCs (I, J), Scgb1a1 and Scgb3a2 for club cells (K, L), and Muc5ac and Spdef for goblet cells (M, N). The mean relative to control cells and the SD as error bars are plotted for each lineage marker. **(O, P)** Representative images for control (O) and ATF4-KD cells (P) processed for immunofluorescence to evaluate club cells (positive for Scgb1a1). The scale bar in (O, P) represents 20 μm. *P*-values for relative gene expression experiments were obtained using a two-tailed *t* test (**** represents *P* < 0.0001, and n.s. means no significative differences).

markers indicative of BSCs, exhibited similar expression patterns in both experimental conditions (Fig 1G and H). Moreover, the expression levels of *Foxj1* and *Mcidas*, markers for MCCs, were comparable between ATF4-KD and LUC-KD cells (Fig 1I and J).

However, upon assessing SC markers, we observed a noteworthy reduction in the expression of *Scgb1a1* and *Scgb3a2* (representing CCs), as well as *Muc5ac* and *Spdef* (corresponding to goblet cells), in ATF4-KD cells compared with control cells (Fig 1K–N).

To further validate these findings, we performed immunofluorescence staining on control and ATF4-KD airway epithelial monolayers using antibodies against p63, Foxj1, and Scgb1a1. Confocal imaging and analysis revealed a similar number of cells labeled with p63 and Foxj1 in both ATF4-KD and LUC-KD cells (Fig S2E–H). However, we observed a reduced number of Scgb1a1-positive cells in ATF4-KD cells (20.4% Scgb1a1-positive cells per field of view) compared with the control conditions (5.8% Scgb1a1-positive cells per field of view) (Fig 1O and P).

Based on the gene expression levels of four SC markers and Scgb1a1 immunofluorescence, we concluded that the proper expression of Atf4 is required for the differentiation of BSCs into SCs in the mouse airway epithelium.

### The innate immune response transcriptional program is overexpressed in Atf4-deficient airway epithelia

To gain a broader understanding of the function of Atf4 in the airway epithelium, we conducted RNA-seq expression analysis on control and ATF4-KD cells after differentiation in ALI14 (14 d of differentiation). Through this analysis, we identified 180 significantly down-regulated genes and 379 significantly up-regulated genes in the absence of Atf4 (Fig 2A and B and Supplemental Data 1). Notably, among the down-regulated genes, we observed the down-regulation of *Atf4* itself and its transcriptional target *Chac1*. In addition, we observed the up-regulation of *Atf2* in cells lacking Atf4. Importantly, we found that all the classical markers for SCs, including goblet and CCs, were down-regulated in ATF4-KD airway monolayers (Fig 2C).

Subsequently, to delve deeper into the functional implications of ATF4 knockdown (ATF4-KD), we conducted a gene ontology analysis using DAVID. Our analysis unveiled a notable up-regulation of the innate immune response program in ATF4-KD cells compared with control cells, encompassing the response to viruses and the cellular response to interferon (Fig 2D). In contrast, the analysis of down-regulated genes did not exhibit a distinct pathway or category (Fig 2D). To validate these findings, we conducted gene set enrichment analysis (GSEA), which produced similar results indicating the enrichment of the innate immune response (Fig 2E). Furthermore, GSEA indicated that two transcriptional programs known to be crucial for BSC differentiation to SCs and MCCs, namely, the p53 pathway and the IL-6/JAK/STAT3 pathways (Tadokoro et al, 2014; Garrido-Jimenez et al, 2021), were down-regulated in ATF4-KD cells.

To further validate our findings regarding the enrichment of the innate immune response in ATF4-KD cells, we conducted a gene validation study focusing on a specific set associated with this response. This gene set comprises Irf7, Oas1, Oas2, Oas3, Mx2, Ifit1, Ifit2, and Ifit3. Across each of these genes, we observed a significant up-regulation in expression within ATF4-KD cells compared with the control conditions (Fig 2F–M).

Moreover, we performed Western blot (WB) analyses targeting Irf7 and assessed STAT1 phosphorylation levels in both control and ATF4-KD cells. Consistent with an elevated innate immune response pathway, particularly the interferon I pathway, we identified an elevation in Irf7 protein levels and increased phosphorylation of STAT1 in ATF4-KD cells relative to the control (Fig 2N and O).

Our RNA-seq data not only confirmed the role of Atf4 in airway epithelial differentiation but also unveiled a potential novel transcriptional function of ATF4 in regulating the innate immune response in the airway epithelium.

### The deficiency of ATF4 resulted in a delay in cilium assembly but did not affect the functioning of ciliated cells

As already mentioned, previous studies have demonstrated the involvement of p53 expression modulation by Mdm2 in the differentiation of BSCs into SCs and MCCs (Garrido-Jimenez et al, 2021). In addition, Tadokoro et al (2014) found that IL-6 signaling through STAT3 played a crucial role in promoting MCC differentiation (Tadokoro et al, 2014). Building upon these findings, we decided to further investigate MCC differentiation.

Upon analyzing the Reactome category in the GSEA, we observed that two categories, namely, intraflagellar transport and anchoring of basal body to the plasma membrane, were enriched in control cells compared with ATF4-KD cells (Fig 3A and B). To directly assess cilia, we performed staining and confocal imaging of acetylated tubulin in control and ATF4-KD cells. No apparent changes were observed when comparing cilia in control and ATF4-KD MCCs (Fig 3C and D). Subsequently, we prepared airway epithelial monolayers for scanning electron microscopy (SEM), which revealed no major structural differences between control and ATF4-KD cells (Fig 4E and F). However, in the SEM images, we observed a less developed apical membrane in ATF4-KD cells, and SCs were more readily identified in control cells (Fig 4E′ and F′).

In previous studies focusing on Jam3-deficient airway epithelia, we observed mild ciliary phenotypes during differentiation (Mateos-Quiros et al, 2021). Therefore, we decided to investigate MCCs throughout the differentiation process by examining the number of MCCs using the basal body protein centriolin as a marker. Our results showed no significant differences in the number of MCCs between control and ATF4-KD cells at ALI4, ALI5, or ALI6 (Fig 3G). Furthermore, the staining with centriolin also allowed us to categorize MCCs into different types representing different stages of differentiation (Vladar & Stearns, 2007). These stages encompass basal body biogenesis protein synthesis (Stage I), basal body biogenesis (Stage II), migration and docking of basal bodies in the apical membrane (Stage III), and the generation of motile cilia from the basal bodies (Stage IV) (Vladar & Stearns, 2007). Analyzing these categories from ALI4 to ALI6, we observed a slight delay in cilium assembly in ATF4-KD cells. In ALI6, we found ~57% of MCCs categorized as type IV/V in ATF4-KD cells compared with 75% in control cells (Figs 3H and S2I–K). Finally, we directly assessed ciliary beating in control and ATF4-KD airway epithelial monolayers using our previously reported assay (Mateos-Quiros et al, 2021). Importantly, we did not observe any significant functional differences in cilia between ATF4-KD and control cells (Fig 3I).

In summary, our analysis of MCCs led us to conclude that although Atf4 deficiency down-regulates the transcriptional program involved in cilium assembly, we only observed a mild delay in cilium formation. Importantly, this delay does not affect cilium function in ATF4-KD airway epithelia.

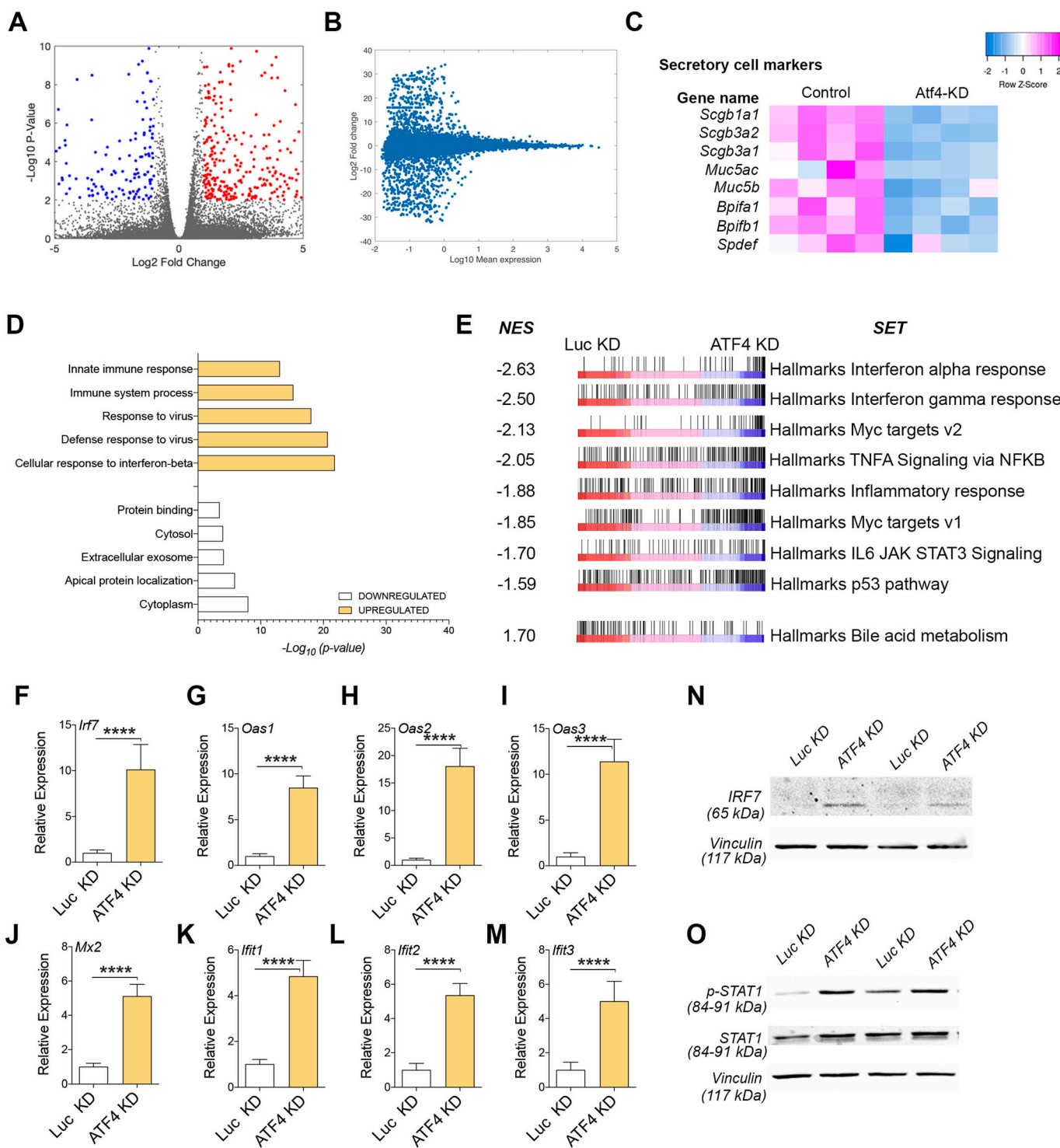

**Figure 2. The innate immune response transcriptional program is overexpressed in Atf4-deficient airway epithelia.**
**(A)** Volcano plot for the RNA-seq (control vs Atf4-deficient). **(B)** MA plot for the RNA-seq (control vs Atf4-deficient). **(C)** Heatmap for secretory cell marker genes in control and ATF4-KD airway epithelia. **(D)** Gene ontology analyses using DAVID. **(E)** Gene set enrichment analysis in control (Luc-KD) versus ATF4-KD cells. **(F, G, H, I, J, K, L, M)** mRNA expression levels in Luc versus ATF4-KD for a selected gene list related to the innate immune response. The mean relative to control cells and the SD as error bars are plotted for each lineage marker. **(N, O)** Western blots for IRF7 (N), p-STAT1 (O), STAT1 (O), and vinculin (as a loading control) to determine protein expression and phospho-STAT1 levels in differentiated MTECs infected with control viruses (Luc-KD) or Atf4-shRNAs (ATF4-KD).

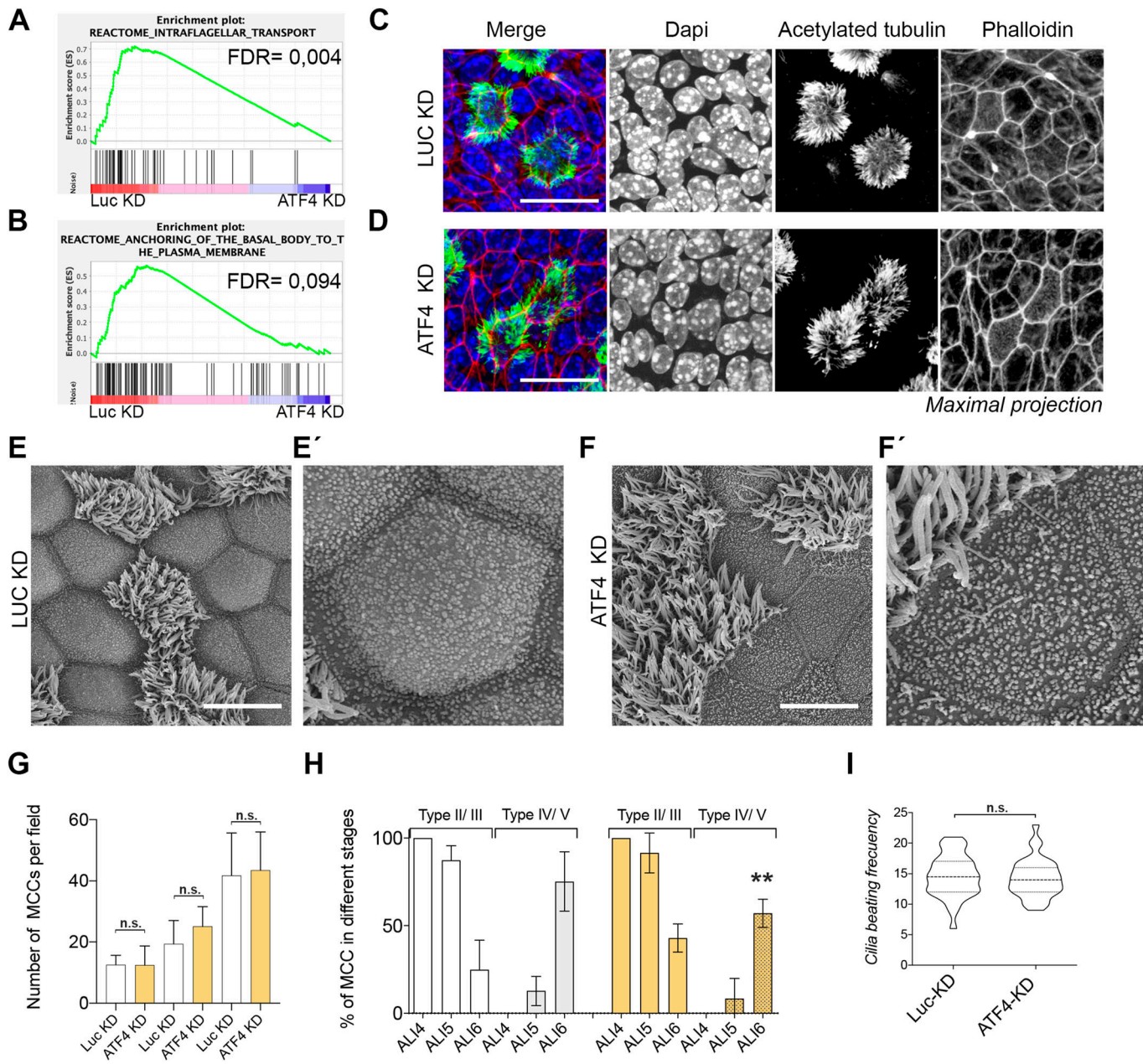

**Figure 3. Atf4 deficiency provoked a delay in cilium assembly but does not affect ciliary cell functioning.**
**(A, B)** GSEA of control versus ATF4-KD cells looking at cilium-related categories. **(C, D)** Maximal projection of confocal images for acetylated tubulin (in green), phalloidin (in red), and nucleus (in blue) in control (C) and ATF4-KD (D) cells. **(E, F)** Scanning electron microscopy images of airway epithelial control (E and close view in E') and ATF4-KD (F and close view in F') cells. **(G)** Quantification of the total number of deuterosomal/MCCs during the initial stages of differentiation. **(H)** Quantification of deuterosomal/MCCs in different stages of differentiation in control and ATF4-KD cells. Type II/III cells are those cells with centriolin staining in aggregates, whereas type IV/V cells are those cells with centriolin staining that disperse at the apical membrane. **(I)** Ciliary beating frequency quantification as number of beats per second in control and ATF4-KD cells (number of measurements, n = 32). Scale bars in (C, D) represent 20 $\mu$m. Scale bars in (E, F) represent 10 $\mu$m. $P$-values in all conditions were obtained using a two-tailed $t$ test (** represents $P < 0.01$, and n.s. means no significative differences).

## PERK/Atf4 signaling is required for SC differentiation

Atf4 is a TF that is activated upon phosphorylation of eIF2$\alpha$ at the Ser-51 residue. Phosphorylation of eIF2$\alpha$ can be mediated by various kinases, including PKR-like ER kinase (PERK), double-stranded RNA-dependent protein kinase (PKR), heme-regulated eIF2$\alpha$ kinase (HRI), and general control non-derepressible 2 (GCN2), each activated by specific cellular stresses (Pakos-Zebrucka et al, 2016). Considering our observation of dysregulated defense against viruses and protein secretion pathways in the RNA-seq data, we aimed to investigate whether PKR or PERK could be responsible for the transcriptional activation of Atf4 during BSC differentiation.

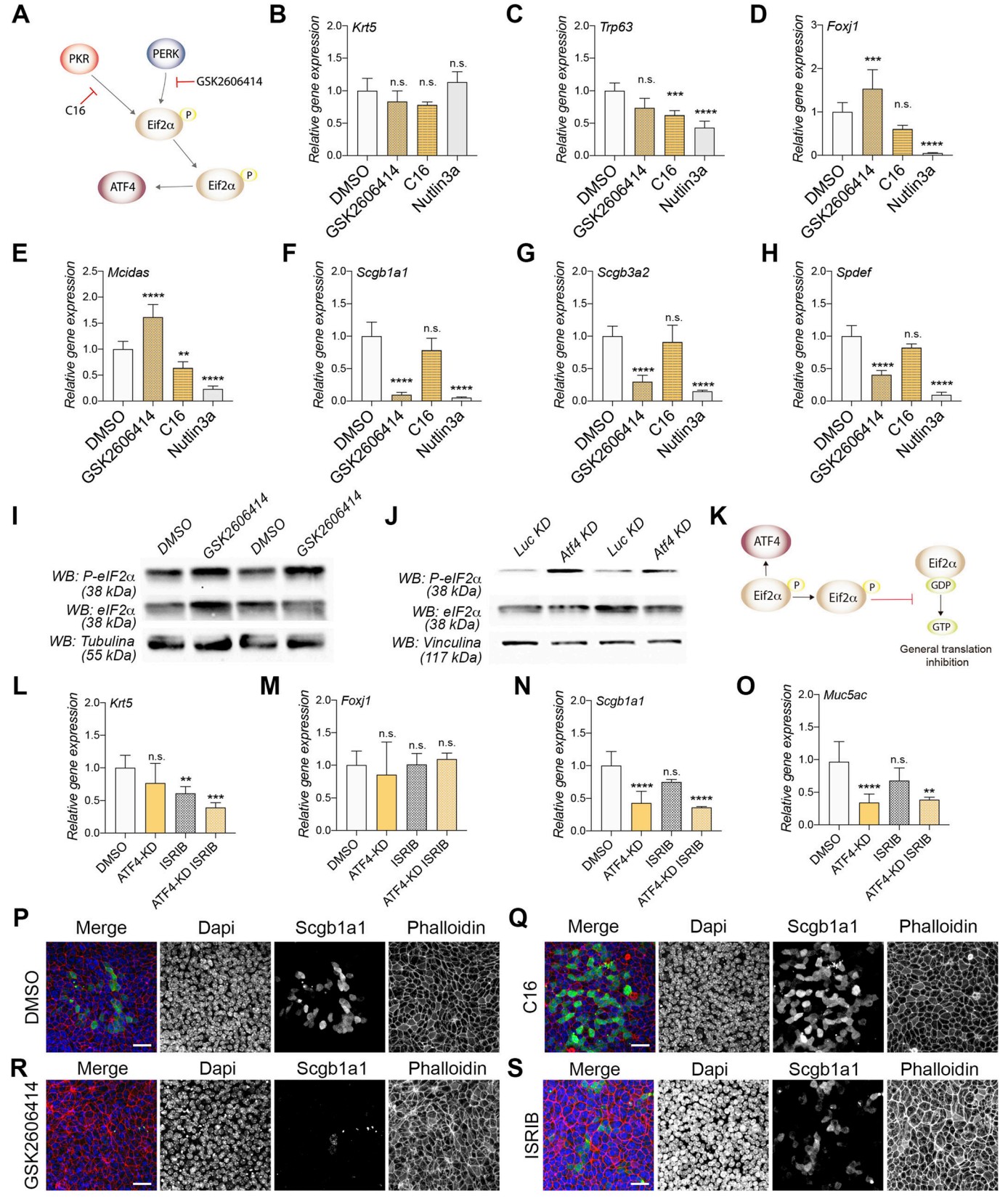

To address this, we employed a pharmacological approach using the PERK inhibitor GSK2606414 and the PKR inhibitor C16 (Fig 4A). Nutlin3a was used as a positive control, and DMSO served as a negative control. After treatment for 14 days during differentiation, we assessed the mRNA expression levels of BSC, MCC, and SC markers. Our findings revealed that the PKR inhibitor C16 did not consistently impact the expression levels of the tested markers (Fig 4B–H). In contrast, treatment with the PERK inhibitor GSK2606414 resulted in a significant decrease in the expression of all three SC markers compared with the DMSO treatment (Fig 4B–H). Notably, this PERK inhibitor phenotype differed from that observed with Nutlin3a, where up-regulation of p53, as previously reported, leads to the suppression of SC and MCC markers (Fig 4B–H).

To further investigate the upstream Atf4 signaling pathway, we examined the expression and phosphorylation levels of eIF2$\alpha$ at serine 51 in ATF4-KD cells and cells treated with the PERK inhibitor. In our experimental conditions, BSCs treated with GSK2606414 during differentiation showed increased phosphorylation of eIF2$\alpha$ compared with DMSO-treated cells (Fig 4I). Similarly, ATF4-KD cells also exhibited elevated levels of phosphorylated eIF2$\alpha$ (Fig 4J). Phosphorylated eIF2$\alpha$ (P-eIF2$\alpha$) functions to repress general translation by inhibiting eIF2B, the guanine nucleotide exchange factor responsible for recycling inactive eIF2 to its active form (Sudhakar et al, 2000). Based on this information, it is possible that the inhibition of general translation in ATF4-KD cells or PERK inhibitor–treated cells could contribute to the observed differentiation phenotype. To explore this possibility, we used ISRIB, a small-molecule inhibitor of the ISR pathway. ISRIB is known to rescue general translation by facilitating the assembly of eIF2B in its active form, even in the presence of phosphorylated eIF2$\alpha$ (Zyryanova et al, 2018; Rabouw et al, 2019). Therefore, we treated control and ATF4-KD cells with ISRIB and analyzed the differentiation status of BSCs. We assessed the expression of *Krt5*, *Foxj1*, *Scgb1a1*, and *Muc5ac* and found that ISRIB treatment did not significantly affect any of the markers examined, except for a mild decrease in *Krt5* (Fig 4L and M). Furthermore, the combination of ATF4-KD and ISRIB treatment did not alter the ATF4-KD phenotype, where down-regulation of SC markers was observed (Fig 4L and M). These results suggest that the differentiation phenotype associated with ATF4 deficiency is not rescued by ISRIB treatment, indicating that other factors or pathways downstream of Atf4 may be involved in this process.

To further validate our findings, we performed immunostaining for p63, Scgb1a1, and Foxj1 in airway monolayers treated with DMSO, C16, GSK2606414, and ISRIB. Consistent with our expectations, we did not observe significant changes in p63 or Foxj1 staining between the DMSO-, C16-, ISRIB-, and GSK2606414-treated cells (Figs 4P–S and S3A–H). However, we noticed that GSK2606414 treatment almost completely abolished the staining of Scgb1a1 in the airway epithelium compared with DMSO or C16 treatments (Fig 4, compare panels P and R). These results provide further support for the crucial role of PERK/Atf4 signaling in the differentiation of SCs, independent of general translation.

It is widely recognized that the ISR and unfolded protein response (UPR), including PERK and Atf4, play crucial roles in cellular adaptation and homeostasis during stress conditions. Although a transient and short-lived ISR/UPR is considered a protective response to resolve stress and restore balance, a prolonged ISR/UPR can lead to cell death. In recent years, it has become increasingly evident that the ISR/UPR also plays important roles during physiological processes, including cell differentiation. SCs, such as pancreatic $\beta$-cells and oligodendrocytes, have been shown to activate the ISR/UPR pathway as part of their differentiation program (Pakos-Zebrucka et al, 2016). The rationale behind this is that in SCs, the ER stress pathways aid in accommodating the increased demands of the secretory pathway, which involves substantial protein synthesis and secretion. For instance, pancreatic $\beta$ cells are capable of synthesizing and secreting a high number of insulin molecules per minute. Our results collectively suggest that Atf4 and the ISR/UPR pathway play important roles in airway epithelial cell differentiation of SCs, which might be related to the ER stress that could be generated during differentiation from BSCs (no-secretory) to SCs.

## ATF4 directly binds to transcriptional regulatory elements of cell differentiation–related genes important for airway epithelial development

Regarding the airway epithelium, Atf4 has been associated with pathological conditions of the respiratory tract. There are three notable examples of this relationship: (i) transcriptional targets of Atf4 are up-regulated in patients with chronic obstructive pulmonary disease (Steiling et al, 2013), (ii) Atf4 expression in the airway epithelium is increased during viral infections, including those caused by SARS-CoV-2 (Vanderheiden et al, 2020), and (iii) the UPR pathway, involving Atf4 and PERK, is activated in a rodent model of chronic lung injury. Interestingly, in this experimental model, mice with reduced Atf4 expression (Atf4+/−) were protected against lung injury (Aggarwal et al, 2018). Based on this information and our data on differentiation, we decided to explore the transcriptional targets responsible for our Atf4 phenotypes.

To accomplish this, we investigated the transcriptional regulatory role of Atf4 during the differentiation of airway cells. We

**Figure 4. PERK/ATF4 signaling is required for secretory cell differentiation.**
**(A)** Schematic representation of the canonical signaling pathway involving Atf4 and the corresponding pharmacological approach employed. **(B, C, D, E, F, G, H)** mRNA expression levels in DMSO-, GSK2606414-, C16-, and Nutlin3a-treated airway epithelial cells. **(C, D, E, G, H)** Krt5 and Trp63 for basal cells (C, D), Foxj1 and Mcidas for MCCs (D, E), Scgb1a1 and Scgb3a2 for club cells (G), and Spdef for goblet cells (H). The mean relative to control cells and the SD as error bars are plotted for each lineage marker. **(I, J)** Western blot images for eIF2$\alpha$, phosphorylated eIF2$\alpha$, and vinculin in DMSO- or GSK2606414-treated cells (I) and control (Luc-KD and ATF4-KD) cells (J). **(L, M, N, O)** mRNA expression levels in DMSO-, ATF4-KD–, ISRIB-, and ATF4-KD + ISRIB–treated cells, including Krt5 (L), Foxj1 (M), Scgb1a1 (N), and Muc5ac (O). **(P, Q, R, S)** Confocal images for Scgb1a1 (in green), phalloidin (in red), and nucleus (in blue) in DMSO (P), C16 (Q), GSK2606414 (R), and ISRIB (S) airway epithelial monolayers treated from ALI0 to ALI14. The scale bar in (P, Q, R, S) represents 20 μm. P-values in all conditions were obtained using a two-tailed $t$ test (**** represents $P < 0.0001$, *** represents $P < 0.001$, ** represents $P < 0.01$, and n.s. means no significative differences).

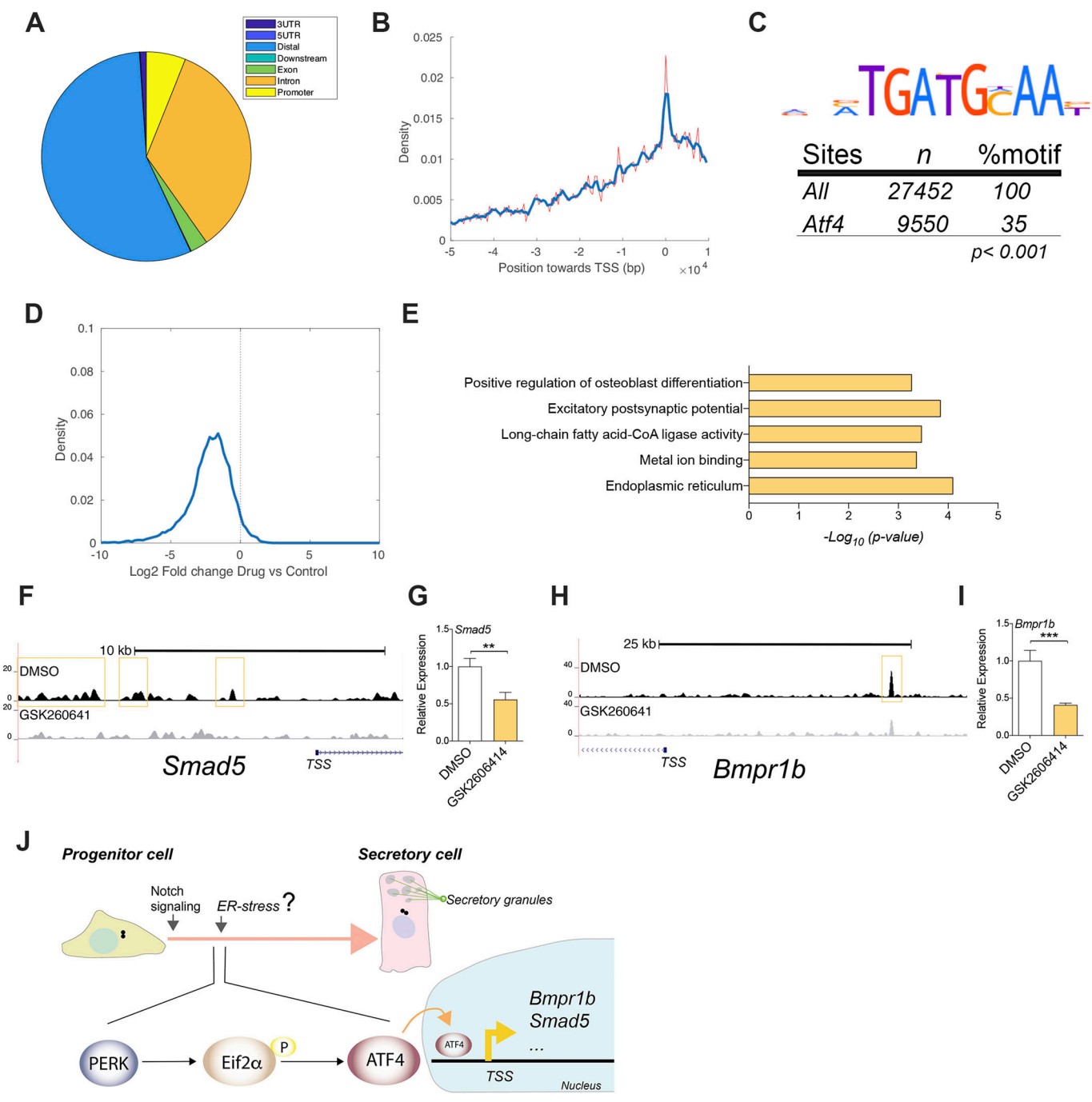

**Figure 5. Atf4 binding during early differentiation is related to the described osteoblast differentiation pathways.**
**(A)** Atf4 peaks position (in base pairs) around transcriptional start sites of known genes. In red and blue, it is represented the raw and smoothed signal, respectively.
**(B)** Distribution of Atf4 peaks in different genomic regions. **(C)** Logo of the Atf4-detected motif in the peak regions, with the number of regions and % of the peaks. **(D)** Comparison of read depth in the Atf4 peak regions in control and drug-treated samples, represented as a density histogram. **(E)** Gene ontology terms overrepresented in the set of genes located nearest to Atf4 peaks, quantified as $-\log_{10}(P\text{-value})$ from Fisher's test. **(F)** Genomic track of aligned reads in the *Smad5* promoter region for Atf4 ChIP-seq in DMSO (top)- and GSK2606414 (bottom)-treated ALI6 MTECs. The yellow box indicates an Atf4 peak detected by MACS2. **(G)** mRNA expression levels in DMSO or GSK2606414 for *Smad5*. **(H)** Genomic track of aligned reads in the *Bmpr1b* promoter region for Atf4 ChIP-seq in DMSO (top)- and GSK2606414 (bottom)-treated ALI6 MTECs. The yellow box indicates an Atf4 peak detected by MACS2. **(I)** mRNA expression levels in DMSO or GSK2606414 for *Bmpr1b*. **(J)** Schematic representation showing the PERK/Atf4 function during airway differentiation. *P*-values in (G, I) were obtained using a two-tailed *t* test (*** represents $P < 0.001$, ** represents $P < 0.01$, and n.s. means no significative differences).

conducted Atf4 chromatin immunoprecipitation sequencing (ChIP-seq) during the early stages of differentiation in ALI6 cell cultures. Our findings revealed that Atf4 was bound to various

sites located in distal, promoter, and intronic regions (Fig 5A). ~30% of these regions contained a canonical Atf4 site, indicating the specificity of the ChIP-seq results (Fig 5C). Despite their

preference for distal regions, the peaks exhibited the highest density near the transcriptional start site of the genes (Fig 5B). Subsequently, we performed ChIP-seq on ALI6 MTEC samples treated with GSK2606414, allowing us to compare them with the untreated samples. The reads within the previously identified Atf4 peaks were significantly altered after drug treatment, indicating that GSK2606414 induced the release of Atf4 from its binding sites (Fig 5D). To conduct gene ontology analysis, we selected the top candidate genes potentially regulated by Atf4 (refer to the Materials and Methods section). Notably, we observed enrichment of genes associated with the ER and osteoblast differentiation (Fig 5E).

Osteoblast differentiation involves the transformation of immature osteoblasts into mature osteoblasts, which are responsible for synthesizing the bone extracellular matrix during osteogenesis, such as collagen type 1 alpha 1 (Col1α1) (Blair et al, 2017). In addition, osteoblasts have been shown to secrete factors such as osteopontin, osteocalcin, and bone morphogenetic proteins (BMPs), which play crucial roles in bone mineralization, calcium homeostasis, and cell signaling pathways involved in osteogenesis (Blair et al, 2017). Based on this knowledge, we selected the candidate genes from the osteoblast differentiation category because of their predicted functional relation to Atf4 (Fig S4A), and analyzed their Atf4 ChIP tracks and their expression after GSK2606414 (Figs 5F–L and S4B–G). We observed that both Smad5 and Bmpr1b decreased their expression after GSK2606414 treatment, concomitant to the release of Atf4 from their promoter regions (Fig 5F–L).

The direct involvement of Smad5 and Bmpr1b in SC differentiation remains uncertain (Wu et al, 2016). Our findings suggest a potential direct connection between Atf4 and successful SC differentiation, mediated in part by Bmpr1b and Smad5 (Fig 5J). Furthermore, although Atf4 and Runx2 have been widely recognized for their significant roles in osteoblast differentiation, as demonstrated in studies by Dobreva et al and Xiao et al, we did not uncover direct control of Runx2 expression by PERK/Atf4 in the context of SC differentiation. These previous studies provide strong evidence for the involvement of Atf4 and Runx2 in osteoblast differentiation, but their direct influence on airway SCs remains to be elucidated (Xiao et al, 2005; Dobreva et al, 2006).

In conclusion, our study provides insights into the initial transcriptional regulatory mechanisms orchestrated by Atf4 during the differentiation of club and goblet cells in the airway epithelium (Fig 5J). By identifying the binding sites of ATF4 and observing their changes after drug treatment, as well as their downstream impact on gene expression, we enhance our comprehension of the complex molecular processes involved in the differentiation of SCs. These findings may have implications for understanding the cellular transition from a non-secretory state to acquiring the primary secretory function.

# Materials and Methods

### In silico screening for candidate TFs in airway differentiation

1,652 mouse proteins with annotated GO:0043565 (sequence-specific DNA binding) were retrieved, and their expression in a scRNA-seq experiment was obtained from the accession code GSE102580 (Plasschaert et al, 2018). Then, we selected the top 100 expressed genes and found their functional interactors by STRING (Szklarczyk et al, 2019). We limited the maximum number of interactors that one candidate could have to 250. Then, we evaluated the mean expression of each gene in either basal or non-basal cells, obtaining the fold change expression per interactor. The average of this fold change for all the interactors of a TF is its combined score. In the case of the aggregated score, the process is the same but using positive (expression > 0) cells for an interactor instead of the mean expression. Fold change is calculated as the ratio between positive non-basal and basal cells, and finally, the average fold change for all the interactors of a TF is obtained. P-values are estimated using random genes as interactors, and by generating the scores.

### Isolation of MTECs

The isolation of MTECs was carried out using a modified version of the procedure described by You et al (2002) in wild-type C57BL/6J adult mice. All animal studies have been performed in accordance with the National and European legislation (Spanish Royal Decree RD53/2013 and EU Directive 86/609/CEE as modified by 2003/65/CE, respectively) and in accordance with the Institute of Laboratory Animal Resources (ILAR) for the protection of animals used for research. Experimental protocols were approved by the Bioethics Committee for Animal Experimentation of the University of Extremadura (Registry July 7, 2017). The mice were euthanized, and the tracheas were carefully dissected from the bronchial main to the larynx and placed in cold Ham's F-12 medium supplemented with penicillin and streptomycin. After removal of vascular fatty tissues and muscle, the clean tracheas were longitudinally excised and incubated with 1.5 mg/ml of pronase (Roche Molecular Biochemicals) in Ham's F-12 medium with penicillin–streptomycin at 4°C for 16 h. Subsequently, FBS (Gibco) was added to achieve a final concentration of 10%. The processed tracheas were discarded, and the isolated cells were collected by centrifugation at 500g for 5 min at 4°C. The cells were then incubated in F-12 medium containing 0.5 mg/ml pancreatic DNase I (Sigma-Aldrich) for 10 min and collected again by centrifugation. Then, cells were seeded within PneumaCult-Ex Plus complete medium (StemCell) in primary tissue culture plates (Corning) for 4 h in 5% $CO_2$ at 37°C to remove fibroblasts. Finally, the supernatant was collected and cells were seeded in 60-mm plates previously treated with type I rat tail collagen (Gibco). Each 60-mm plate was seeded with cells from three tracheas.

### Culture of mouse tracheal cells and treatments

MTECs were cultured in PneumaCult-Ex Plus medium (StemCell) in 5% at 37°C, and this medium was replaced every 2 d until 70–80% confluence. After this first step, cells were detached from plates by two consecutive incubations, with 0.02% EDTA in PBS for 5 min at 37°C, and Accutase (Gibco) for 5 min at RT. MTECs were seeded at a confluence of $9 \times 10^4$ cells/cm2 in the polyester porous membrane (Transwell 0.4-$\mu$m pores; Corning), and upper and lower chambers were filled with PneumaCult-Ex Plus, every two days. When porous

membranes were at confluence (4–6 d), cells were switched to air–liquid interface (ALI) in the lower chamber for differentiation. ALI medium was maintained until the end of the differentiation for 14 d (ALI14), and it was removed every two days. Samples in ALI2, ALI4, ALI7, and ALI14 were processed for ARN, protein, or any other assay. For PERK, PKR, and eIF2α inhibition were used GSK2606414 (5 µM, #5107; TOCRIS), C16 (500 nM, #5284; TOCRIS), and ISRIB (10 µM, #5382; TOCRIS), respectively. These inhibitors were added to the medium during differentiation of MTEC cultures, and DMSO was used as a vehicle control.

## Transepithelial resistance measurements of airway epithelial monolayers

The permeability of tight junctions during the differentiation process of MTECs was assessed by measuring transepithelial resistance (TEER) every 2 d until ALI14. The procedure involved washing the upper chamber of transwells once with PBS with calcium and magnesium. Subsequently, DMEM at 37°C was added to both the upper and lower chambers of the transwells, and transepithelial resistance was measured using EVOM3. The electrode was positioned in the upper chamber to measure the resistance, which was recorded in kiloohms (kΩ).

## Culture of mouse airway organoids (AOs)

To culture mouse airway organoids (AOs), the expanded MTECs were prepared as described previously. When the MTECs reached 60–70% confluence, they were detached and prepared for organoid formation. 96-well plates were coated with 100 µl of 30% Matrigel in PneumaCult-Ex Plus medium and incubated at 37°C for 20 min until gelation occurred. Then, 500 MTEC control or ATF4-KD cells were suspended in 100 µl of 2% Matrigel in PneumaCult-Ex Plus and were seeded onto the Matrigel layer. After 5 d, images of the airway organoids were captured using EVOS FLoid Cell Imaging Station (Invitrogen).

## Lentivirus production and infection of airway epithelial cells

To generate short hairpin RNAs (shRNAs) against Atf4, the pLKO.1 vector was used. This vector uses the U6 human promoter to drive the expression of shRNAs and contains a puromycin resistance selection sequence. The shRNA against luciferase was obtained from Miguel Fidalgo's laboratory. The shRNAs (Table S1) were cloned into the pLKO.1 vector using EcoRI and AgeI sites, following the protocol described by Woo et al (2019). The purified DNA, along with the packaging and envelope vectors psPAX2 and pMD2.G, was transfected into HEK-293T cells using polyethylenimine (PEI). After 48 and 72 h, the viral supernatants were concentrated using Amicon Ultra-15 filters by centrifugation. The approximate concentration of infectious virus particles was adjusted to $2 \times 10^5$ ml. For transduction, 75 µl of concentrated virus was added to 2 ml of medium containing 8 µg/ml of polybrene in a 60-mm plate. MTECs were selected after infection by treating them with puromycin at a final concentration of 3 µM for 48 h before proceeding with the experiments.

## Gene expression analysis

Total RNA was isolated using the illustra RNAspin Mini kit (GE Healthcare). After elution, 200–400 ng of RNA was subjected to reverse transcription using High-Capacity cDNA Reverse Transcription Kit (Applied Biosystems) according to the manufacturer's instructions. Quantitative PCR (qPCR) was performed to analyze gene expression using the oligonucleotides listed in Table S2 and Power SYBR Green PCR Master Mix (Applied Biosystems) as per the provided instructions. Melting curve analyses were conducted to validate the specificity of the PCRs. For RNA-seq analysis, libraries from RNA samples were built following Illumina recommendations at Novogene UK, and then sequenced. Raw reads were pseudo-aligned to the *Mus musculus* (mm10) genome using kallisto (Bray et al, 2016). Negative binomial distribution was used for obtaining differentially expressed genes, and ontology differences were retrieved using DAVID (Sherman et al, 2022). An ad hoc script transformed these outputs to the specific format required to be analyzed by GSEA (Subramanian et al, 2005). Raw and processed data can be accessed at Figshare (https://figshare.com/s/46e4f215d41323965927).

## Immunofluorescence of airway epithelial monolayers

MTEC cultures differentiated at ALI14 were fixed in 4% formaldehyde (PolyScience) for 10 min at RT. Subsequently, the samples were permeabilized with 0.1% Triton for 10 min and blocked with 2% BSA (Roche) for 45 min. Primary antibodies were incubated overnight at 4°C in the blocking buffer. The primary antibodies used for immunofluorescence (IF) were anti-p63 (#ab124762, 1:200; Abcam), anti-Foxj1 (#14-9965-82, 1:200; Invitrogen), anti-SCGB1a1 (#sc-9772, 1:100; Santa Cruz), and anti-acetylated tubulin (#T6793; Sigma-Aldrich). In contrast, MTEC cultures differentiated at ALI4, ALI5, and ALI6 were fixed in cold 99.8% methanol (Panreac) for 10 min at –20°C. The next steps were similar to those fixed in formaldehyde, with the exception that the primary antibody used for this immunofluorescence (IF) was centriolin C9 (#sc-365521, 1:100; Santa Cruz).

After the incubation with primary antibodies, the samples were washed five times in 0.1% Triton and then incubated for 1 h with fluorescent secondary antibodies diluted in the same buffer. The secondary antibodies used for IF were Alexa Fluor 594 anti-goat (#A11058, 1:200; Invitrogen), Alexa Fluor 594 anti-rabbit (#A11012, 1:200; Invitrogen), Alexa Fluor 488 anti-rabbit (#A21206, 1:200; Invitrogen), and Alexa Fluor 594 anti-mouse (#A11005, 1:200; Invitrogen). Nuclei were stained with DAPI at a concentration of 0.5 µg/ml (#62248; Thermo Fisher Scientific). Five additional washes were performed in 0.1% Triton. Slides were mounted in Vectashield (Vector Labs), and images were captured using an Olympus FV1000 confocal microscope. After acquisition, the images were processed using ImageJ (Fiji) and Adobe Photoshop CC 2018.

For the quantitative assessment of cells expressing a specific cell-type marker, we counted the number of positive cells in a minimum of four fields of view captured using a confocal microscope and then projected them into a single image. To determine the percentage of positive cells, DAPI staining was employed to calculate the total number of cells in a given field of view.

### Western blot (WB) analysis

Cell lysis was performed using ice-cold lysis buffer composed of 50 mM Tris–HCl (pH 7.5), 1 mM EGTA, 1 mM EDTA, 1 mM sodium orthovanadate, 5 mM sodium pyrophosphate, 10 mM sodium fluoride, 0.27 M sucrose, 0.1 mM phenylmethylsulfonyl fluoride, 0.1% (vol/vol) 2-mercaptoethanol, 1% (vol/vol) Triton X-100, and complete protease inhibitor cocktail (Roche). The protein concentration was determined using the Bio-Rad protein assay, and 10 µg of protein was subjected to SDS–PAGE and transferred onto nitrocellulose membranes (Bio-Rad Laboratories). The membranes were then blocked with 5% dry milk in Tris-buffered saline containing 0.05% Tween-20 and incubated overnight at 4°C with primary antibodies. The primary antibodies for Western blotting were diluted in blocking solution. The antibodies used were anti-ATF4 (#11815, 1:100; Cell Signaling), anti-IRF7 (#4920, 1:1,000; Cell Signaling), anti-STAT1 (#14994, 1:1,000; Cell Signaling), anti-p-STAT1 (#8826, 1:1,000; Cell Signaling), anti-p-eIF2$\alpha$ (#3398, 1:1,000; Cell Signaling), anti-eIF2$\alpha$ (#5324, 1:1,000; Cell Signaling), anti-vinculin (#V4505, 1:1,000; Sigma-Aldrich), and anti-$\alpha$-tubulin (#32-2500, 1:1,000; Invitrogen) in blocking solution. After several washes with TBS/Tween, the membranes were incubated with HRP-conjugated secondary antibodies (anti-rabbit-HRP, #7074; Cell Signaling, and anti-mouse-HRP, #7076; Cell Signaling) in blocking solution (1:1,000) for 1 h at RT. After additional washes with TBS/Tween, the proteins were visualized using a chemiluminescence detection system (SuperSignal West Dura; Thermo Fisher Scientific) and detected with iBright CL1000.

### SEM of airway epithelial monolayers

Differentiated MTECs at ALI14 were prepared for SEM by fixing them with 2.5% glutaraldehyde for 90 min at 4°C. They were then washed in 0.2 M cacodylate buffer and subsequently stained with 1% osmium tetroxide (Sigma-Aldrich) in 0.2 M cacodylate for 2 h at 4°C. Samples were further dehydrated by incubation in increasing concentrations of ethanol (10%, 30%, 50%, 70%, 90%, and 100%) for 20 min each. Afterward, the samples were dried using the technique of liquid carbon dioxide critical point, followed by gold sputter coating. Finally, the samples were visualized using a Quanta 3D FEG (ESEM-FIB; FEI Company) electron microscope.

### Ciliary beating frequency measurement in MCCs

Ciliary beating frequency was assessed in ALI14 MTEC control or ATF4-KD cells. The upper chamber of 12-mm transwells was incubated with 5 µl of Dynabeads Streptavidin C1 (Invitrogen) in 300 µl of $Ca^{2+}/Mg^{2+}$-PBS at 37°C for 10 min. After incubation, the medium from the upper chamber was removed, and the transwells were incubated at 37°C for 90 min. Before recording, 100 µl of $Ca^{2+}/Mg^{2+}$-PBS at 37°C was added on top of the transwell.

Ciliary beating movies were captured using a Motic AE20 microscope and an iPhone X at 240 frames per second. A custom MATLAB function was developed to analyze the movement of the beads and quantify the ciliary beating frequencies using fast Fourier transform analysis. This allowed for the assessment and quantification of ciliary beating frequencies based on the movement of the beads in the recorded videos.

### ChIP and massive sequencing

ChIP to analyze Atf4 binding to different promoters was performed essentially as described previously (Mulero-Navarro et al, 2006; Roman et al, 2008). Then, libraries for ChIP-seq were built following Illumina recommendations at STAB VIDA. 20M reads were obtained per input and IP sample, respectively. They were aligned to the *M. musculus* (mm10) genome using BWA, and peaks were retrieved with MACS2 (Feng et al, 2012). ChIPseeker (Wang et al, 2022) and HOMER (Heinz et al, 2010) were used to annotate the peaks, and custom MATLAB functions were developed for other analyses. Candidate genes were obtained from peaks with a canonical Atf4 site ($P < 0.0001$) and located between +0.5 and −20 kb from the transcription start site. Gene ontology was analyzed by DAVID (Sherman et al, 2022). Raw and processed data are available at Figshare (https://figshare.com/s/46e4f215d41323965927).

### Statistical analyses

Data analysis was performed using two-tailed $t$ tests to compare the control conditions with various experimental groups. GraphPad Prism software was used for this analysis.

## Data Availability

Data from NGS experiments are publicly available at the Figshare server (https://figshare.com/s/46e4f215d41323965927).

## Supplementary Information

## Acknowledgements

Confocal microscopy and scanning electron microscopy were performed at the UEX microscopy core facilities. This work was supported by BFU2017-85547-P, TED2021-130560B-I00, and PID2021-126905NB-I00 grants from the Ministry of Economy, IB18014 from Junta de Extremadura to JM Carvajal-Gonzalez, and GR21140 from Junta de Extremadura to S Mulero-Navarro. S Garrido-Jimenez was a recipient of a Fellowship from the Universidad de Extremadura. S Diaz-Chamorro and CM Mateos-Quiros were recipients of a Fellowship from Junta de Extremadura. All Spanish funding is co-sponsored by the European Union FEDER program.

### Author Contributions

JF Barrera-Lopez: conceptualization, investigation, and methodology.
G Cumplido-Laso: conceptualization, investigation, methodology, and writing—original draft, review, and editing.
M Olivera-Gomez: investigation and methodology.
S Garrido-Jimenez: investigation and methodology.

S Diaz-Chamorro: investigation and methodology.

CM Mateos-Quiros: investigation and methodology.

DA Benitez: formal analysis and investigation.

F Centeno: investigation.

S Mulero-Navarro: formal analysis, investigation, and writing—original draft, review, and editing.

AC Roman: conceptualization, software, investigation, and writing—original draft.

JM Carvajal-Gonzalez: conceptualization, formal analysis, supervision, funding acquisition, and writing—original draft, review, and editing.

## Conflict of Interest Statement

The authors declare that they have no conflict of interest.

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
