## [Reviewer comments · Life Science Alliance]

Life Science Alliance

Early Atf4 activity Drives Airway Club & Goblet Cell Differentiation

Juan Barrera-Lopez, Guadalupe Cumplido-Laso, Marcos Olivera-Gomez, Sergio Garrido-Jimenez, Selene Diaz-Chamorro, Clara Mateos-Quiros, Dixan Benitez, Francisco Centeno, Sonia Mulero-Navarro, Angel-Carlos Roman, and Jose Carvajal-Gonzalez

DOI: <https://doi.org/10.26508/lsa.202302284>

Corresponding author(s): Jose Carvajal-Gonzalez, University of Extremadura

Review Timeline:	Submission Date:	2023-07-19
	Editorial Decision:	2023-09-06
	Revision Received:	2023-11-26
	Editorial Decision:	2023-12-18
	Revision Received:	2023-12-28
	Accepted:	2023-12-28

Transaction Report:

September 6, 2023

Re: Life Science Alliance manuscript #LSA-2023-02284-T

Dr. Jose Maria Carvajal-Gonzalez
Universidad de Extremadura
Departamento de Bioquímica, Biología Molecular y Genética

Dear Dr. Carvajal-Gonzalez,

Thank you for submitting your manuscript entitled "Early Atf4 transcriptional activity enables the differentiation of club and goblet cells in the airway epithelium" to Life Science Alliance. The manuscript was assessed by expert reviewers, whose comments are appended to this letter. We invite you to submit a revised manuscript addressing the Reviewer comments.

Thank you for this interesting contribution to Life Science Alliance. We are looking forward to receiving your revised manuscript.

Sincerely,

B. MANUSCRIPT ORGANIZATION AND FORMATTING:

Reviewer #1 (Comments to the Authors (Required)):

This manuscript by Dr. Barrera-Lopez and colleagues addresses the PERK/ATF4 axis in controlling differentiation of airway basal cells. Through a series of studies, they demonstrate that ATF4 is important for the differentiation of basal to club and goblet cells. Their manuscript is elegant in its use of primary airway basal cell cultures, differentiation in ALI and their use of complementary approaches that include EM and ciliary beating studies, along with computational and ChIP-seq approaches. Their overall conclusions that ATF4 plays a physiological role in allowing secretory cells to cope with an enhanced burden of secretory protein processing/folding provides important new insights into the cellular processes that govern epithelial plasticity. I have the following comments:

Figure 1A involves the re-analysis of a publicly available data set to study expression differences and analyze potential interactors and the overall premise of this manuscript is based on this analysis. The accession code for this should be included in the methods and figure legend. How did the authors ensure that basal cells, ciliated cells and secretory cells were properly annotated?

Did the authors determine whether ATAC seq databases exist in conjunction with scRNA seq data. This is the more direct, and perhaps state of the art route to link gene expression changes with chromatin structure and putative transcription factor occupancy, provided that these data sets exist for airway basal cells?

Can the authors quantify the proportion of cells staining for SCGB1A1 in Figure 1N and O?

It would be helpful to include a panel of the most significant gene hits following ATF4 knock down, including hits from the GSEA. Otherwise the data is limited to the volcano and MA plots without any additional information.

The ultrastructural changes in ATF4 KD cells shown in Figure 3 E and F are quite interesting but are poorly explained and not quantified. The scale bars are missing. The authors should provide more quantitative detail to these structural differences.

Minor comments:

Proper nomenclature should be followed with abbreviations for proteins showing in all capital letters.

The manuscript notably figures should be checked for typographical errors. Numerous errors are apparent including the following: Figure 3C should read acetylated tubulin, The Y axis description of Figure 3H should be corrected. The legends in Figure 5 are mislabeled

Reviewer #2 (Comments to the Authors (Required)):

The authors of the manuscript Early Atf4 transcriptional activity enables the differentiation of club and goblet cells in the airway epithelium use a model of cultures mouse tracheal epithelial cells to show that this transcription factor has a role in development of secretory cells in ALI cultures. Using Atf4 shRNA knockdown and pharmacologic PERK inhibition, they demonstrate that mouse ALI epithelial cultures differentiate relatively normally into ciliated epithelial cells in the absence of Atf4/PERK axis but secretory cells are specifically impacted. ChIP seq identifies localization of Atf4 binding to relevant genes in secretory cell development.

Comment on main conclusions from the paper (Major comments)

1. Atf4 expression is essential for differentiation of secretory cells: Overall the main conclusions of this section are supported by data.

a. Supplementary Fig 2, Line 245: The paper states that a reduced number of positive cells are seen but there is no quantitation of cell numbers with positive staining for Scgb1a1. Wording should be changed to reflect the qualitative nature of the current conclusion or proportion of positive stained cells in sections should be quantitative. The latter would be preferred if possible (Figure 1N, 1O).

b. Figure 1D: Is any protein-level validation available for knockdown efficiency?

c. For all figures in the paper, is it not clear of the number of replicates. The N of tracheas per dish to begin is 3. However, are multiple different starting populations used? How many replicate wells are within each experiment? This information would be

needed in the methods and/or the figure legends.

2. Innate immune response transcriptional program is overexpressed in Atf4 deficiency airway epithelia: Relatively well supported by data.

a. Line 257: The timing of when the collection of BSC is performed for this analysis is not clear from the text or the figures.

b. While the pathway analysis for innate immune response genes is suggestive of a target, and supports previous data, the conclusions would be much more strongly supported if some type of validation of RNA-seq data was performed (presentation of individual gene RNA-seq data plus a validation of a subset of these genes by qPCR). In its current state, the nature of the pathways identified is limited as the use of gene ontology signatures can be highly biased. Are all ISG and pro-inflammatory cytokines impacted or is there a more distinct pattern?

3. Deficiency of Atf4 delays cilia assembly but not function: reasonably well supported by data

a. Line 297: What is the basis for saying no significant changes were observed comparing cilia in control to Atf4-KO MCC? Was any quantitation performed?

b. Line 949: What does n=32 represent? Number of ALI cultures? It is not clear.

c. Supplementary figure 2: Quantitation of the FoxJ1 staining of proportion of positive cells between groups would be helpful. From looking, it appears there is a difference in amount of positive staining between the two representative images.

d. Supplementary figure 2: Figure legend does not match figures.

4. PERK/Atf4 signaling is required for secretory cell differentiation

a. Figure 4: It seems the effect of PERK inhibitor has more effect on MCC than the Atf4 knockdown. Commentary on this by the authors should be expanded. Does this suggest an additional role for either of these proteins independent of the PERK-ATF4 axis?

5. Atf4 directly binds to transcriptional regulatory elements of cell differentiation related genes: Conclusions are supported adequately.

Minor comments

A. Methods: Please add dilutions for all antibodies used in IF and WB

B. Line 910: Typo on Atf4

C. Line 1103: Order of genes is different in figures than figure legend

D. Is any cross-sectional histology performed to evaluate these cultures?

Reviewer #1 (Comments to the Authors (Required)):

This manuscript by Dr. Barrera-Lopez and colleagues addresses the PERK/ATF4 axis in controlling differentiation of airway basal cells. Through a series of studies, they demonstrate that ATF4 is important for the differentiation of basal to club and goblet cells. Their manuscript is elegant in its use of primary airway basal cell cultures, differentiation in ALI and their use of complementary approaches that include EM and ciliary beating studies, along with computational and ChIP-seq approaches. Their overall conclusions that ATF4 plays a physiological role in allowing secretory cells to cope with an enhanced burden of secretory protein processing/folding provides important new insights into the cellular processes that govern epithelial plasticity. I have the following comments:

We thank the reviewer for his/her positive assessment of our work.

Figure 1A involves the re-analysis of a publicly available data set to study expression differences and analyze potential interactors and the overall premise of this manuscript is based on this analysis. The accession code for this should be included in the methods and figure legend. How did the authors ensure that basal cells, ciliated cells and secretory cells were properly annotated?

Did the authors determine whether ATAC seq databases exist in conjunction with scRNA seq data. This is the more direct, and perhaps state of the art route to link gene expression changes with chromatin structure and putative transcription factor occupancy, provided that these data sets exist for airway basal cells?

Following the reviewer suggestion, we have added the accession code (GSE102580) in methods and figure 1 legend.

Can the authors quantify the proportion of cells staining for SCGB1A1 in Figure 1N and O?

We have quantified the percentage of Scgb1a1 positive cells in control and ATF4-KD conditions. We have also extended the "Immunofluorescence of airway epithelial monolayers" method section to explain how we did those quantification.

It would be helpful to include a panel of the most significant gene hits following ATF4 knock down, including hits from the GSEA. Otherwise, the data is limited to the volcano and MA plots without any additional information.

Following the reviewer suggestion, we have added a file (Supplementary file 1) with upregulated and downregulated transcripts including their fold-change and p-value.

The ultrastructural changes in ATF4 KD cells shown in Figure 3 E and F are quite interesting but are poorly explained and not quantified. The scale bars are missing. The authors should provide more quantitative detail to these structural differences.

We have incorporated Figure 3 E and F missing error bars.

Minor comments:

Proper nomenclature should be followed with abbreviations for proteins showing in all capital letters.

The manuscript notably figures should be checked for typographical errors. Numerous errors are apparent including the following: Figure 3C should read acetylated tubulin, The Y axis description of Figure 3H should be corrected. The legends in Figure 5 are mislabeled

We have carefully revised our manuscript, paying special attention to correcting typographical errors. Additionally, we have rectified errors in the axis and legend.

Reviewer #2 (Comments to the Authors (Required)):

The authors of the manuscript Early Atf4 transcriptional activity enables the differentiation of club and goblet cells in the airway epithelium use a model of cultures mouse tracheal epithelial cells to show that this transcription factor has a role in development of secretory cells in ALI cultures. Using Atf4 shRNA knockdown and pharmacologic PERK inhibition, they demonstrate that mouse ALI epithelial cultures differentiate relatively normally into ciliated epithelial cells in the absence of Atf4/PERK axis but secretory cells are specifically impacted. ChIP seq identifies localization of Atf4 binding to relevant genes in secretory cell development.

We thank the reviewer for his/her positive assessment of our work.

Comment on main conclusions from the paper (Major comments)

1. Atf4 expression is essential for differentiation of secretory cells: Overall the main conclusions of this section are supported by data.

a. Supplementary Fig 2, Line 245: The paper states that a reduced number of positive cells are seen but there is no quantitation of cell numbers with positive staining for Scgb1a1. Wording should be changed to reflect the qualitative nature of the current conclusion or proportion of positive stained cells in sections should be quantitative. The latter would be preferred if possible (Figure 1N, 1O).

We have quantified the percentage of Scgb1a1 positive cells in control and ATF4-KD conditions. We have also extended the “Immunofluorescence of airway epithelial monolayers” method section to explain how we did those quantifications.

b. Figure 1D: Is any protein-level validation available for knockdown efficiency?

We have assessed two antibodies targeted against mouse ATF4, leading to the identification of one that effectively functions. In the most recent iteration of our manuscript, we have incorporated a Western blot (WB) for ATF4 alongside an appropriate loading control. As a result, we have adjusted Figure 1, its corresponding legend, the WB method section, and the main text. These revisions were made to rectify panel references and ensure the inclusion of essential information.

c. For all figures in the paper, is it not clear of the number of replicates. The N of tracheas per dish to begin is 3. However, are multiple different starting populations used? How many replicate wells are within each experiment? This information would be needed in the methods and/or the figure legends.

Allow us to provide a more detailed explanation of our experimental procedures. We began by freezing vials containing basal stem cells (BSCs) obtained from tracheas. To generate each batch of BSCs, we utilized 20 mice and harvested 20 tracheas, which we combined to create a pool of approximately 10 to 20 vials of BSCs. These vials were then stored at -80°C or in liquid nitrogen.

For each experiment, we selected one or multiple vials from a specific batch of BSCs. Consequently, for each experiment, we initiated with BSCs that had been cultured in expansion media to attain a sufficient quantity of BSCs for subsequent steps in the differentiation protocol. Our experiments were conducted a minimum of four times, ensuring that we had a minimum of four independent differentiations.

2. Innate immune response transcriptional program is overexpressed in *Atf4* deficiency airway epithelia: Relatively well supported by data.

a. Line 257: The timing of when the collection of BSC is performed for this analysis is not clear from the text or the figures.

We apologize for any previous lack of clarity in our explanation of this particular aspect.

We have changed the text to:

*“To gain a broader understanding of the function of *Atf4* in the airway epithelium, we conducted RNA-seq expression analysis on control and ATF4-KD cells after differentiation in ALI14 (14 days of differentiation).”*

b. While the pathway analysis for innate immune response genes is suggestive of a target, and supports previous data, the conclusions would be much more strongly supported if some type of validation of RNA-seq data was performed (presentation of individual gene RNA-seq data plus a validation of a subset of these genes by qPCR). In its current state, the nature of the pathways identified is limited as the use of gene ontology signatures can be highly biased. Are all ISG and pro-inflammatory cytokines impacted or is there a more distinct pattern?

In response to the reviewer's suggestion, we have conducted a validation of a set of genes associated with the innate immune response. This set includes *Irf7*, *Oas1*, *Oas2*,

Oas3, Mx2, Ifit1, Ifit2, and Ifit3. Furthermore, we conducted Western blot (WB) analyses for Irf7 and assessed STAT1 phosphorylation levels. In each of these cases, we observed a noteworthy increase in activation in ATF4-KD cells when compared to the control conditions. These data are now in main Figure 2.

We are sincerely grateful for this specific reviewer's comment, as it has significantly enhanced the quality and clarity of our conclusions in this particular aspect.

3. Deficiency of Atf4 delays cilia assembly but not function: reasonably well supported by data

a. Line 297: What is the basis for saying no significant changes were observed comparing cilia in control to Atf4-KO MCC? Was any quantitation performed?

Based on this comment, we have chosen to enhance the precision of our interpretation of these images. As we did not quantify our visual data results, we have decided to modify the text accordingly.

Previous line 297:

"No significant changes were observed when comparing cilia in control and ATF4-KD multiciliated cells (Figure 3, panels C and D)"

Current version 301:

"No apparent changes were observed when comparing cilia in control and ATF4-KD multiciliated cells (Figure 3, panels C and D)"

b. Line 949: What does n=32 represent? Number of ALI cultures? It is not clear.

We apologize for the lack of clarity. When we mentioned, n=32 means, it indicates that we have recorded the beating of 32 cells for each condition, collected from three distinct cultures.

c. Supplementary figure 2: Quantitation of the FoxJ1 staining of proportion of positive cells between groups would be helpful. From looking, it appears there is a difference in amount of positive staining between the two representative images.

We have conducted an analysis to determine the proportion of multiciliated cells (MCC) in the specified experimental conditions. Our findings indicate that approximately 5-6% of the cells within each field of view are MCC, both in the control and ATF4-KD monolayers.

d. Supplementary figure 2: Figure legend does not match figures.

We apologize for this mistake. We have now corrected the text to match the content of Supplementary Figure 2.

4. PERK/Atf4 signaling is required for secretory cell differentiation

a. Figure 4: It seems the effect of PERK inhibitor has more effect on MCC than the Atf4 knockdown. Commentary on this by the authors should be expanded. Does this suggest an additional role for either of these proteins independent of the PERK-ATF4 axis?

At this juncture, we aim to refrain from excessive speculation regarding this difference, given that we are comparing a gene knockdown scenario with a drug treatment. Consequently, while the reviewer posits the possibility of an additional role for PERK, it could also signify residual Atf4 function or cellular adaptations in response to chronic Atf4 knockdown as opposed to acute drug treatment. If required, we can introduce these concepts sporadically throughout the text, although doing so might compromise clarity.

5. Atf4 directly binds to transcriptional regulatory elements of cell differentiation related genes: Conclusions are supported adequately.

We thank the reviewer for this comment.

Minor comments

A. Methods: Please add dilutions for all antibodies used in IF and WB

We have included the dilution for each used antibody.

B. Line 910: Typo on Atf4

We have fixed this mistake.

C. Line 1103: Order of genes is different in figures than figure legend.

We have fixed the text.

D. Is any cross-sectional histology performed to evaluate these cultures?

We did not perform cross sectional histology.

December 18, 2023

RE: Life Science Alliance Manuscript #LSA-2023-02284-TR

Dr. Jose Maria Carvajal-Gonzalez
University of Extremadura
Departamento de Bioquímica, Biología Molecular y Genética
Avenida de Elvas SN
Badajoz, Badajoz 06011
Spain

Dear Dr. Carvajal-Gonzalez,

Thank you for submitting your revised manuscript entitled "Early Atf4 activity Drives Airway Club & Goblet Cell Differentiation". We would be happy to publish your paper in Life Science Alliance pending final revisions necessary to meet our formatting guidelines.

- please add a Category for your manuscript in our system
- please add the Twitter handle of your host institute/organization as well as your own or/and one of the authors in our system
- the full name (first name, middle name as initial, last name) of each author should be given on the title page
- please add a conflict of interest statement to your main manuscript text
- please add an Author Contributions section to your main manuscript text
- please add your main, supplementary, and table legends to the main manuscript text after the references section
- please upload all figure files as individual ones, including the supplementary figure files; all figure legends should only appear in the main manuscript file
- please upload your Tables in editable .doc or excel format; tables should be numbered consecutively with Arabic numerals (1, 2, 3, 4). They can be included at the bottom of the main manuscript file or sent as separate files.
- there are call-outs for figure 5K-M, and this figure doesn't have these panels, please correct
- please add callouts for Figures S3A-H and S4A-G to your main manuscript text
- approval for mouse work is needed. Please indicated this in the Materials and Methods section, and who granted the approval.
- ChIP-seq data should be publicly shared, and the details of how to access it should be mentioned in a Data Availability statement. The current figshare reference does not result in a match on figshare. Upload to a database is recommended.

Figure Checks:

- please add sizes next to each blot
- please make sure to include scale bars and their sizes on each set of microscopy images

A. FINAL FILES:

- An editable version of the final text (.DOC or .DOCX) is needed for copyediting (no PDFs).
- High-resolution figure, supplementary figure and video files uploaded as individual files: See our detailed guidelines for preparing your production-ready images, <https://www.life-science-alliance.org/authors>

B. MANUSCRIPT ORGANIZATION AND FORMATTING:

Sincerely,

Reviewer #1 (Comments to the Authors (Required)):

The authors have addressed my prior concerns in a thoughtful manner. I have no further concerns. This manuscript provides new insights into airway basal cell differentiation.

Reviewer #2 (Comments to the Authors (Required)):

The authors of the manuscript Early Atf4 transcriptional activity enables the differentiation of club and goblet cells in the airway epithelium use a model of cultures mouse tracheal epithelial cells to show that this transcription factor has a role in development of secretory cells in ALI cultures. Using Atf4 shRNA knockdown and pharmacologic PERK inhibition, they demonstrate that mouse ALI epithelial cultures differentiate relatively normally into ciliated epithelial cells in the absence of Atf4/PERK axis but secretory cells are specifically impacted. ChIP seq identifies localization of Atf4 binding to relevant genes in secretory cell development.

The authors have successfully addressed all of my comments from the previous review.

December 28, 2023

RE: Life Science Alliance Manuscript #LSA-2023-02284-TRR

Dr. Jose Maria Carvajal-Gonzalez
University of Extremadura
Departamento de Bioquímica, Biología Molecular y Genética
Avenida de Elvas SN
Badajoz, Badajoz 06011
Spain

Dear Dr. Carvajal-Gonzalez,

Thank you for submitting your Research Article entitled "Early Atf4 activity Drives Airway Club & Goblet Cell Differentiation". It is a pleasure to let you know that your manuscript is now accepted for publication in Life Science Alliance. Congratulations on this interesting work.

DISTRIBUTION OF MATERIALS:

Again, congratulations on a very nice paper. I hope you found the review process to be constructive and are pleased with how the manuscript was handled editorially. We look forward to future exciting submissions from your lab.

Sincerely,
